# Does an Environmental Management System Affect Green Inno-Vation: The Role of Green Financing in China's Tourism Sector in a Circular Economy

**Xiang Ji [1], Shiqi Zhang [2] and Yuan Lu [1,]***

[1] School of Economics, Hainan University, Haikou 570228, China
[2] HNU-ASU Joint International Tourism College, Hainan University, Haikou 570228, China
* Correspondence: luyuan_vip@outlook.com

**Abstract:** The occurrence of climate change is becoming a challenge for the survival of business organizations. These changes pressure business organizations to adopt strategies to protect the natural environment. In order to cope with these changes, business organizations concentrate on strategic decisions regarding the protection of the natural environment due to the demand of various stakeholders. Continuous and updated information on environmental issues is required to successfully formulate and implement decisions to protect the natural environment. Therefore, the environmental management system (EMS) is an important mechanism that enables business organization to collect information about the demands of various stakeholders regarding the natural environment. Most studies have examined the green innovation (GI) of the tourism sector and related the innovation of natural environmental protection activities with management's capabilities and strategic decisions. Limited studieshave considered EMSs as important to bring GI into the tourism sector. GI comprises various environmental initiatives that play a vital role in impacting the GI of the tourism sector worldwide. However, these are ignored by researchers. Therefore, we examined the effect of the EMS on GI. Moreover, we also examined the extent to which green financing of the tourism sector intervenes in the EMS and GI link. Data were collected from 322 managers in the tourism sector. The collected data were analyzed with the help of correlation and regression techniques. The study findings confirmed that the EMS positively affects GI, while green financing mediates the connection between the EMS and GI. Hence, this study offers numerous practical suggestions for improving the GI of the tourism sector in the emerging circular economy.

**Keywords:** environmental management system; green financing; green innovation; tourism sector





## 1. Introduction

The appearance of climate change exerts pressure on business organizations to implement major technological, product, and organizational innovations to meet the demands of various stakeholders and fulfill regulatory requirements [1]. Changes in the natural environment exert pressure on business organizations to cope and respond to the demands of the various stakeholders regarding the protection of the natural environment. Climate changes such as global warming, pollution pervasiveness, and decreasing levels of natural resources cause massive changes in the operations of business organizations [2]. The survival of business organizations is primarily based on the inclusion of decisions about protecting the natural environment in their planning. In this regard, principles of the circular economy have a significant role through which business organizations cope with the underlying circumstances of the natural environment [3] and include green innovation (GI) as an important element in the strategic decision.

Existing studies documented various environmental policies to improve the natural environment [3]. Existing studies have extensively considered GI; however, these studies ignore its pre-conditions and determinants. The view of GI is significantly deliberated in

the existing studies; however, these studies consider few determinants and pre-conditions for improving GI. The last few decades have witnessed unprecedented climatic changes, pervasive pollution, and abnormal depletion of resources at a daunting speed [4]. Climate changes force organizations to develop practices and systems that ensure involvement in protecting the natural environment [2]. Businesses have focused on formulating and implementing novel methods to cope with environmental changes [3]. The EMS is one of the important determinants of GI, which plays an essential role in protecting nature. GIs in the hotel and tourism industry majorly depend on their involvement in corporate social responsibility and protecting the natural environment. However, hardly any studies show the impacts of EMSs and green financing to improve the GIs of hotels and the tourism industry. The current study has been conducted to fulfill this research gap by investigating the defining role of EMSs and green financing in achieving GI. With sound EMS and green financing activities, the tourism sector can maximize its potential to satisfy customers [4].

The tourism industry is highly susceptible to and affected by several internal and external factors. Tourism performance is badly affected by unpredicted disasters, such as health, financial, and natural crises. The tourism and travel industries are more susceptible to these circumstances than other industries [5]. Tourism sectors worldwide are taking significant actions to address environmental problems [6]. Strategies to protect the natural environment have become essential for many organizations. Previous research studies have considerably examined GIs in the context of ISO 14001 certification [7], senior perceptions of green information systems [8], board environmental committees [9], green HRM [10], etc. However, deliberation about its determinants and pre-conditions, especially in the context of the tourism sector, still exists. The EMS has emerged as a key predictor of organization GIs [11].

The EMS is concerned with organized policies and initiatives of business organizations for the environment that facilitate implementing environmentally friendly strategies to improve GIs [12]. Multiple environmental systems exist that stimulate the tourism industry to initiate programs to improve the environment [13]. Among these plans, green financing is important for implementing green activities that improve the GI of the tourism sector [14]. Green financing investments in tourism industries are critical in improving GI [15]. Through green financing, tourism industries can provide environmentally friendly services to tourists to improve GI. Therefore, we studied the mediating role of green financing between EMSs and GI.

Green innovation reduces environmental burdens and deals with the required changes necessary to protect the natural environment [12]. Through innovative activities, tourism management increases operational efficiency by adopting new technologies to improve the natural environment [15]. Green innovation is concerned with innovation activities that minimize environmental damage and provide safe services to tourists [14]. With the help of green innovation, the tourism industry develops environmentally friendly services most effectively [16].

This study identifies the connection between an EMS, green financing, and green innovation. Furthermore, we analyzed the direct impact of an EMS on GI. Moreover, the current study also examined the mediating role of green financing between the EMS and GI. This research has meaningful implications for tourism and hospitality management due to their response to the various tourist environmental demands.

*1.1. EMS and GI of Tourism*

The tourism industry's performance is mainly linked to the prevailing conditions of a particular economy. Existing studies highlighted that economic, environmental, financial, political, and cultural conditions are the primary determinants of tourism performance [16]. Most researchers empirically documented that high-growth economies with stable political and environmental conditions positively impact the performance of the tourism industry [17]. The EMS is a mechanism through which an organization can efficiently deal with the risks to the natural environment [18]. In other words, we can say that the EMS allows an

organization to identify and control the influence of its products, services, and operational activities on the natural environment [19].

Daddi et al. [20] defined EMS as an understandable and organized process of recommending and executing environmental policies. Environmental public relations efforts are one dimension of the EMS based on a good relationship between an organization and its stakeholders [21]. Simply, we can say that environmental public relations efforts are concerned with the interaction between an organization, its customers, and local communities. Leonldou et al. [22] suggested resources and outcomes of EMSs and suggested that relationship building is an important key resource of an EMS. Relationship building involves a firm's ability to express friendly networks with external stakeholders. Organizations can easily understand stakeholders' demands and respond accordingly through these relationships, enhancing overall innovation activities. On the other hand, the environmental policy and training dimension of the EMS is based on environmental organization-related policies to assess their contribution towards the natural environment and arrange workshops for employees for waste reduction [23].

The existing literature has documented that EMSs are key to improving GI [12]. To improve GI, many organizations have greatly emphasized environmental policies such as preventing pollution and minimizing their waste and emissions [15]. On the other hand, through environmental public relations efforts, dimensions of EMS organizations increase relationships with their stakeholders, such as customers and local communities [20], who demand products and processes that are environmentally acceptable. Organizations respond to stakeholders' demands using novel production methods, harmless material use, and controlling pollution to improve the natural environment [14]. The level of GI increases when organizations are more inclined to develop EMSs and adopt environmental resource conservation efforts and environmental public relations efforts. Therefore, we formulated that:

**Hypothesis H1.** *EMSs positively predict the GI of tourism.*

### 1.2. Mediating Role of Green Financing

Green financing involves investment and financial activities to protect the environment [24]. Environmental issues encourage the management of the tourism industry to exercise the mechanism of green financing as a part of their strategic decisions [25]. To protect the natural environment, green financing has a critical role in improving tourism performance [15]. The green financing mechanism is necessary for all kinds of business organizations to meet the demands of stakeholders regarding the protection of the natural environment. Green financing has gained strategic importance in business due to global warming and material waste [14].

In today's business world, particularly in the tourism industry, stakeholders exert pressure on the inclusion of green financing for business activities [19]. Tourism researchers have documented that green financing significantly predicts tourism [14,15]. Therefore, tourism organizations deal with environmental issues by meeting tourists' demands regarding the green environment. The view of tourism GI was significantly deliberated in the existing studies; however, these researchers considered few determinants and preconditions for the GI of the tourism sector.

Green innovation is embedded in those activities that minimize environmental damage and provide safe services to tourists [26]. Green innovation involves developing required products and services during environmental crises [15]. With the help of green innovation, the tourism industry develops environmentally friendly services that effectively protect individuals [27]. Green innovation reduces environmental burdens, dealing with required changes during times of crisis [17]. Through innovative activities, tourism management increases operational efficiencies by adopting new technologies to reduce the impact of an emerging crisis [26]. Therefore, green innovation is critical in developing required activities, products, and services. Besides giving importance to the environmental aspect,

green innovation allows for collecting information regarding the environmental stance of all stakeholders that increases tourism performance [28]. Green innovation facilitates information gathering to develop green products and services for improving tourism [29]. This valuable improvement in products and services financially supports the tourism sector to improve performance.

The current study examines how green financing interferes with the connection between the EMS and GI. Study Hypothesis 1 explains the role of the EMS in improving GI activities in tourism sectors. On the other hand, we formulated that the positive role of the EMS on the GI link is mediated by green financing. Investments in improving tourism services enable tourism units to maximize GI performance through green financing (Figure 1).

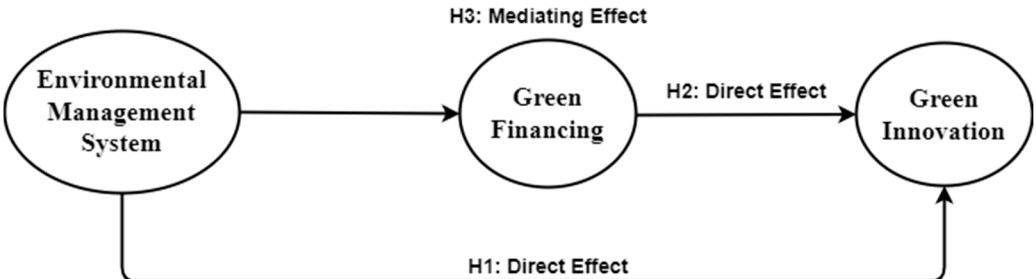

**Figure 1.** Theoretical framework.

**Hypothesis H2.** *Green financing positively predicts the GI of tourism.*

**Hypothesis H3.** *Green financing mediates the connection between the EMS and GI of the tourism sector.*

### 1.3. Methodology

To analyze the collected data and test the study hypotheses, this study was based on a cross-sectional design. Collected data were analyzed with the help of correlation and regression statistical techniques. Research data for the study in hand was collected from managers of the tourism sector. The sample frame consists of all tourism concerns. To achieve the study objectives, we approached tourism units of China in order to elucidate the research objectives. A list of 1929 managers was provided by the representatives of the tourism sector. We selected 322 respondents using simple random sampling. Questionnaires were sent to selected respondents of the tourism sector for their responses on the study constructs. During data collection, only 276 respondents returned the filled questionnaires. After scrutiny, only 235 responses were finalized for analysis.

### 1.4. Study Measures

Table 1 provides information about the respondent's demographics, including the gender, age group, qualification, and working experience of the managers.

The independent variable of EMS was measured based on environmental policies. The construction of the EMS is based on environmental public relations efforts, environmental policy, and training and environmental resource conservation efforts. Table 2 contains the details of the study measures. In the current study, we adapted 32 items from existing studies, e.g., from Choi et al. [30], on the topic of EMSs that are relevant to the context of tourism. Furthermore, mediating variable green financing is measured based on investment activities of the tourism sector to improve the green environment. The items used for the measurement of green financing were adapted from the work of Yasir et al. [31] and Majid et al. [32]; 10 items were adapted for the measurement of green financing. Finally, the dependent variable, green innovation, was measured by 15 items adapted from the work of Chen [33], Cheng and Shiu [34], and Ruzzier et al. [35].

**Table 1.** Respondents' characteristics.

|  |  | N | % |
|---|---|---|---|
| Gender | Male | 293 | 74.94 |
|  | Female | 98 | 25.06 |
| Age (in years) | 22–30 | 70 | 17.90 |
|  | 31–35 | 139 | 35.55 |
|  | 36–40 | 88 | 22.51 |
|  | 41–50 | 75 | 19.18 |
|  | 51–60 | 12 | 3.07 |
|  | Above 60 | 7 | 1.79 |
| Qualification | 12 years | 34 | 8.70 |
|  | 14 years | 107 | 27.37 |
|  | 16 years | 177 | 45.27 |
|  | Above 16 years | 73 | 18.67 |
| Experience | 5–10 years | 29 | 7.42 |
|  | 11–15 years | 109 | 27.88 |
|  | 16–20 years | 78 | 19.95 |
|  | 21–25 years | 45 | 11.51 |
|  | 26–30 years | 61 | 15.60 |
|  | Above 30 years | 69 | 17.65 |

**Table 2.** Measurement scale.

| Variable | Adapted/Adopted/ Self-made | Authors | of Items |
|---|---|---|---|
| Environmental management system | Adapted | Choi et al. [30] | 32 items |
| Green financing | Adapted | Yasir et al. [31] and Majid et al. [32] | 10 items |
| Green innovation | Adapted | Chen [33], Cheng and Shiu [34], and Ruzzier et al. [35] | 15 items |

## 2. Results

In the first step of the analysis, we confirmed the association between constructs. Coefficients of correlation confirmed the connection among the study variables. Table 3 presents the outcomes of the correlation analysis. The findings suggested the direction of the association among the study constructs. The results of the EMS correlation showed a positive relationship with GI (0.39 **) and green financing (0.35 **). Furthermore, green financing positively correlated with green innovation (0.31 **). The correlation findings among study constructs were significant and met the acceptance criteria. Based on these findings, further analysis was conducted to test the study hypotheses. For the purpose of empirical findings of the study hypotheses, we applied regression analysis.

**Table 3.** Correlation statistics.

| Constructs | Mean | SD | 1 | 2 | 3 |
|---|---|---|---|---|---|
| Environmental management system | 3.8 | 0.93 | 1 |  |  |
| Green financing | 3.5 | 0.91 | 0.35 ** | 1 |  |
| Green innovation | 3.6 | 0.90 | 0.39 ** | 0.31 ** | 1 |

Significance level: ** $p < 0.01$.

### 2.1. Constructs Reliability and Validity

Coefficients of correlation confirm the association among study variables. In the second step, we conducted a reliability and validity test for the establishment of the reliability and validity of the constructs. The reliability and validity of the study constructs are established on the basis of Cronbach's Alpha, Factor Loading, Composite Reliability, and Average Variance Extracted (AVE). Table 4 shows the findings of reliability and validity analyses. The calculated results revealed that construct validity and reliability were established because the values generated for reliability and validity were above the threshold. After confirming the correlation, reliability, and validity of the study constructs, we analyzed the collected data to test the study hypotheses.

**Table 4.** Reliability measures.

| Variables | FL | AVE | Cronbach's $\alpha$ | CR |
|---|---|---|---|---|
| Environmental Management System | 0.60–0.85 | 0.679 | 0.85 | 0.82 |
| Green financing | 0.65–0.91 | 0.713 | 0.74 | 0.84 |
| Green Innovation | 0.73–0.90 | 0.756 | 0.83 | 0.88 |

### 2.2. Regression Analysis

After confirming correlation and construct validity, we tested the formulated hypotheses using regression analysis. The regression analysis coefficients confirmed the dependent variables' dependency on the independent variable. Table 5 reports the findings of the regression analysis. Models 1 and 3 present the outcomes of the direct impact of the EMS on GI and green financing.

**Table 5.** Results for the mediating effects of green financing.

| Variables | DV: Green Innovation | | DV: Green Financing |
|---|---|---|---|
| | **Model 1** | **Model 2** | **Model 3** |
| Predictors | | | |
| Environmental management system | 0.22 ** (0.033) | 0.13 (0.092) | 0.32 *** (0.028) |
| Green financing | | 0.36 *** (0.049) | |
| $R^2$ | 0.38 | 0.38 | 0.38 |
| Adjusted $R^2$ | 0.34 | 0.35 | 0.35 |
| *F*-value | 20.65 *** | 49.86 ** | 37.64 *** |
| Durbin-Watson | 2.096 | 2.014 | 2.215 |

Significance level: ** $p < 0.01$; *** $p < 0.001$. Unstandardized coefficients reported. Standard errors in parentheses.

The coefficients in Models 1, 2, and 3 confirmed the positive impact of the EMS on green financing and GI. The findings also suggested that green financing positively affects the GI activities of the tourism sector. The first hypothesis shows the direct impact of the EMS on GI. The results generated for the direct effect of the EMS and GI confirmed that the EMS positively predicts the GI of the tourism sector. The estimate of regression analysis for the EMS and GI are positive and meet the acceptance criteria. The estimated coefficient (0.22 **) confirmed that the EMS directly determined the GI activities of the tourism sector. Therefore, we accept the study H1.

The second hypothesis shows the direct impact of green financing on GI. The estimated coefficient in Table 5 confirms the direct impact of green financing on GI. The estimated coefficient of Model 3 (0.32 **) confirms the direct impact of green financing on GI. Hence, we also accept the study H2.

Study hypothesis three proposed the mediating effect of green financing. In the current study, we formulated that green financing positively mediates the link between the EMS and GI. Model 2 of Table 5 shows the mediating effect of green financing. The findings confirmed that green financing positively mediates the association between the EMS and GI.

After green financing was added to the regression model, the coefficient of the EMS showed an insignificant effect on GI, i.e., 0.13. Meanwhile, the coefficient of green financing for the EMS remained significant, i.e., 0.36 **. The coefficient of Model 2 for the mediating effect is positive and significant and confirms the intervening role of green financing between the EMS and GI. On the basis of these findings, we accept study H3.

## 3. Discussion

The current study contributes to the existing literature in a distinct way and identifies how an EMS affects the GI of the tourism sector. The emergence of global warming in the last few years has become a serious challenge for many organizations concerning their survival and performance. These challenges exert pressure on business organizations to adopt environmentally friendly strategies to protect the natural environment and meet the demands of various stakeholders. The same is the case with the tourism sector regarding the adoption of green strategies.

The current study consists of three hypotheses that explain the association between the EMS, green financing, and green innovation. First, the EMS is a critical factor for organizations to respond to the demands of various stakeholders regarding environmental issues. The EMS plays a foundational role in the formulation of green strategies. EMSs have been utilized to establish the environmental stance, thinking, and motivation of organizations worldwide, which is considered a critical factor in improving the GI of the tourism sector. Organizations with sound EMSs can collect valuable information necessary to formulate green strategies. EMSs provide information regarding the various demands of stakeholders toward environmental changes. Previous findings support an EMS as a significant predictor of GI [11,12].

The study H2 formulated that green financing positively predicts GI. Business organizations achieve environmental performance through proper investment to protect the natural environment. Green financing allows business organizations to adopt environmental-related strategies necessary to protect the natural environment. Investment in green activities ensures the formulation of green business strategies for business organizations [22,36]. The tourism sector can create innovative ideas for promoting the natural environment through green financing. Therefore, organizations with green financing activities are more likely to engage in GI activities [14,15]. GI facilitates the adoption of new ways to deal with tourists and protect the natural environment. Green financing supports the implementation of new technologies for the safety of tourists. Green financing provides a foundation for implementing GI in the tourism sector. The study's findings confirmed the positive effect of green financing on GI.

Finally, the third hypothesis supports that green financing mediates between the EMS and GI, through which the tourism industry uses precautionary measures to protect the natural environment. The findings suggested that the EMS provides updated information regarding the demands of various stakeholders about environmental issues. Using this information, business organizations formulate green business strategies and make investments for the execution of formulated strategies [37,38]. Green financing plays a critical role in the implementation of green strategies. Through green financing, the tourism sector implements the required measures to cope with the demands of the natural environment [39]. The tourism sector cannot correctly perform its green innovation activities without green financing [40]. Green financing facilitates implementing GI activities formulated with the help of EMSs in the tourism industry. The findings of the current studies regarding the study H2 align with existing studies [14,36].

### 3.1. Theoretical Contribution

This study significantly and theoretically contributes to the existing literature on environmental performance. This study also significantly adds to the existing literature on EMSs. The main contribution is formulating the model that tested the EMS as a foundational factor of the tourism sector's GI. Limited studies have considered environmental management

factors to improve the green performance of tourism. Firstly, a comprehensive research model was developed for the tourism sector, testing both the direct effect of the EMS and the mediating effect of green financing for implementing formulated environmental strategies and green innovation activities to protect the natural environment.

Furthermore, the current study also adds to the existing body of knowledge by explaining the role of green financing in improving tourism environmental performance. The current study's findings suggested that through proper investment in promoting and protecting the natural environment, the tourism sector meets the demands of tourists and enhances environmental performance. Green financing is an important mechanism for the thriving formulation of new models and activities for tourism GI activities [11]. Existing studies in the relevant field ignore the role of green financing concerning its combination with an EMS. Therefore, the current study considered this aspect, fills this research gap, and focuses on green financing as a potential mediator of the EMS and GI link.

### 3.2. Practical Implications

The findings of the current study provide practical implications. First, the findings suggested that the tourism sector can improve environmental performance during global warming with the help of green financing and green innovation. By doing so, adopting green innovation is possible when these tourism units optimistically react to the environmental demands of stakeholders, especially tourists. Second, this study suggests that the tourism sector's green financing activities increased environmental performance. Therefore, through an organized EMS, proper green financing, and green innovation, the tourism industry can improve the natural environment and strengthen the sector's performance.

### 4. Conclusions

The aim of the study was to highlight the foundational role of the EMS in green business strategies in tourism. We formulated green business strategies based on information regarding the natural environment. The current study explained that an EMS generates valuable information and is foundational in promoting a tourism management stance toward protecting the natural environment. Through an EMS, the tourism sector can build relationships with potential stakeholders. Based on these relationships, tourism management can respond according to the influence of products, services, and operational activities on the natural environment. In the current study, we formulated that the EMS is one of the foundational factors facilitating the formulation of green strategies in the tourism sector. The most constructive aspect of tourism GI is green financing under global warming. The findings revealed that an EMS increase GI. Furthermore, the findings suggest that green financing mediates between the EMS and GI activities in tourism.

**Author Contributions:** All three authors (X.J., S.Z. and Y.L.) contributed equally. All authors have read and agreed to the published version of the manuscript.

**Funding:** This paper is funded by one of the phased results of the general project of the National Social Science Foundation of China in 2020, "Research on Statistical Accounting andDynamic Monitoring of Regional Tourism under Multi-Source Data Fusion" (Project No.20BTJ031).

**Institutional Review Board Statement:** The study was conducted in accordance with theDeclaration of national tourism and approved by the Institutional Review Board (or EthicsCommittee) of Hainan University HN-124/66-2022 dated: 13 September 2022.

**Informed Consent Statement:** Informed consent was obtained from all subjects involved in the study.

**Data Availability Statement:** The raw data which support the conclusions of the current article will be made available on demand.

**Conflicts of Interest:** The authors declare no conflict of interest.

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
