# Peer review of "Does an Environmental Management System Affect Green Inno-Vation: The Role of Green Financing in China’s Tourism Sector in a Circular Economy"

_sustainability, doi:10.3390/su15086411_

Round 1

Reviewer 1 Report

Comments to Author/s

Thank you for the opportunity to review this manuscript draft. The topic seems interesting and I hope the author/s will come up will the best possible outcomes.  

· The abstract is nicely written and it properly mirrors the study.

The introduction section was nicely developed and I appreciate the author/s hard work but it needs further effort in terms of clarity and trying to explain in terms of  · sustainability comprehensively. Further, I will suggest author/s that at first give the form of each main concept, i.e. in line number 30, the author/s directly gave GI which is supposed to be green innovation (GI) and later they can give GI. Fix those.    I will suggest author/s start their literature and hypothesis section from section 1.1 onwards. It will look much better. And I will also suggest adding a few more references in this section in order to support your evidence in a more appropriate way. 

The methodology and the Study measures are nicely developed and the author/s are rational regarding what and how they are going to · estimate. Appreciated

The results section is presented well but still, I will suggest author/s that if possible try to include a few more references to support their results. 

The discussion section is satisfactory, nicely developed and divided into proper sections. But I will suggest authors add a few references in this section also. 

I will suggest to authors that their manuscript is very poorly formatted therefore check guidelines for author/s properly and format the manuscript according to the journal temple.

Best of Luck!

Author Response

Dear Editor

Manuscript ID: sustainability-2257229

The paper titled " Does environmental management system affect green innovation: Role of green financing of tourism sector of China in circular economy?" has been reviewed as per comments of the referee(s).These changes are incorporated and highlighted in the text; the details of the changes are as follows:

Reviewer 1 Comments

Reviewer 1 Changes

The abstract is nicely written and it properly mirrors the study.

The introduction section was nicely developed and I appreciate the author/s hard work but it needs further effort in terms of clarity and trying to explain in terms of  · sustainability comprehensively. Further, I will suggest author/s that at first give the form of each main concept, i.e. in line number 30, the author/s directly gave GI which is supposed to be green innovation (GI) and later they can give GI. Fix those.    I will suggest author/s start their literature and hypothesis section from section 1.1 onwards. It will look much better. And I will also suggest adding a few more references in this section in order to support your evidence in a more appropriate way. 

The methodology and the Study measures are nicely developed and the author/s are rational regarding what and how they are going to · estimate. Appreciated

The results section is presented well but still, I will suggest author/s that if possible try to include a few more references to support their results. 

The discussion section is satisfactory, nicely developed and divided into proper sections. But I will suggest authors add a few references in this section also. 

I will suggest to authors that their manuscript is very poorly formatted therefore check guidelines for author/s properly and format the manuscript according to the journal temple.

  1. Accepted, no changes are suggested by the reviewer.

  1. These changes are incorporated as per suggestions of the reviewer in the introduction and literature review section.

  1. Accepted, no changes are suggested by the reviewer.

  1. Results section is rewrite and includes more information in this section by adding new tables.

  1. These changes are incorporated.

  1. These changes are incorporated as per the suggestion and manuscript is formatted as per the journal requirement.

Reviewer 2 Report

The paper is well written, and the results are of interest. I am concerned that hypotheses were analyzed using only one method: Cronbach’s Alpha, Factor Loading, Composite Reliability and Average Variance Extracted (AVE), thus the findings are potentially subject to a model error.  I believe the results of the paper will be more convincing if alternative statistical methods are used leading to the same conclusions.

References:

1.Brian Dennis, José Miguel Ponciano, Mark L. Taper and Subhash R. Lele, Errors in Statistical Inference Under Model Misspecification: Evidence, Hypothesis Testing, and AIC,Front. Ecol. Evol., 21 October 2019,Sec. Environmental Informatics and Remote Sensing

Volume 7 - 2019 | https://doi.org/10.3389/fevo.2019.00372
https://www.frontiersin.org/articles/10.3389/fevo.2019.00372/full

2. Mikko Ronkko and Eunseong Cho,An Updated Guideline for Assessing Discriminant Validity
Organizational Research Methods ª The Author(s) 2020 Article reuse guidelines: sagepub.com/journals-permissions DOI: 10.1177/1094428120968614 journals.sagepub.com/home/orm 2022, Vol. 25(1) 6– 47, https://journals.sagepub.com/doi/pdf/10.1177/1094428120968614

Author Response

Dear Reviewer

Manuscript ID: sustainability-2257229

The paper titled " Does environmental management system affect green innovation: Role of green financing of tourism sector of China in circular economy?" has been reviewed as per comments of the referee(s).These changes are incorporated and highlighted in the text; the details of the changes are as follows:

Reviewer 2 Comments

Reviewer 2 Changes

The paper is well written, and the results are of interest. I am concerned that hypotheses were analyzed using only one method: Cronbach’s Alpha, Factor Loading, Composite Reliability and Average Variance Extracted (AVE), thus the findings are potentially subject to a model error.  I believe the results of the paper will be more convincing if alternative statistical methods are used leading to the same conclusions.

These changes are incorporated as per the suggestions of the reviewer by adding new tables and rewriting the whole section of result. These changes are highlighted in the manuscript.

Reviewer 3 Report

The article has severe flaws that does not allow publication in the current situation.

The abstract is very poor and lacks the main elements to understand the study.

The article would benefit from a native proofreading.

The model is too simple for a structural equations approach. A single regression would be enough.

The hypos are not derived rrom previous, empirical studies, so doubts remain on their appropriateness.

Correlations are too weak to justify the relationships.

In the SEM, the main tests are lacking. Pleae see Hair et al. (2018) for a comprehensive overview on tests..

It lacks a full description on the indicators, a cross-load an weak load analysis providade by EFA.

It lacks a demographic analysis.

One of the constructs are oversized, as CR > 0.95, so at least one indicator should be removed.

Refs are few and some are not relevant.

It lacks a discussion on the implications of the study.

I suggest rewritting the article and a resubmission.

best regards.

Author Response

Dear Reviewer

Manuscript ID: sustainability-2257229

The paper titled " Does environmental management system affect green innovation: Role of green financing of tourism sector of China in circular economy?" has been reviewed as per comments of the referee(s).These changes are incorporated and highlighted in the text; the details of the changes are as follows:

Reviewer 3 Comments

Reviewer 3 Changes

The article has severe flaws that does not allow publication in the current situation.

The abstract is very poor and lacks the main elements to understand the study.

The article would benefit from a native proofreading.

The model is too simple for a structural equations approach. A single regression would be enough.

The hypos are not derived rrom previous, empirical studies, so doubts remain on their appropriateness.

Correlations are too weak to justify the relationships.

\

It lacks a demographic analysis.

One of the constructs are oversized, as CR > 0.95, so at least one indicator should be removed.

It lacks a discussion on the implications of the study.

The abstract of the manuscript is rewrite and include the main elements as per the suggestion.

Native proof reading is done to eliminate the grammatical mistakes.

As per the suggestion the statistical techniques is changed.

These changes are incorporated in the literature review section.

Correlations analysis is run with the inclusion of demographic variables

Demographic information is provided and new tables are added in the result section of the manuscript.

There are some mistakes in the column heading which is corrected in the Table.

These changes are incorporated in the discussion section of the manuscript.

Round 2

Reviewer 2 Report

I recommend the publication in the current (revised) form.

Author Response

Dear Reviewer

Manuscript ID: sustainability-2257229

The paper titled " Does an environmental management system affect green innovation: The role of green financing in China’s tourism sector in a circular economy" has been reviewed as per comments of the referee(s). These changes are incorporated and highlighted in the text; the details of the changes are as follows:

Reviewer 1 Comments

Reviewer 1 Changes

Accepted for Publication

Thanks for the acceptance of required changes

Reviewer 3 Report

The English remains poor, there are parts that resembles a naive use of language. Please hire a high-level ghost-writer to improve your text.

Demographic variabled can not take part of correlation analisis. Please remove then and redo the analysis.

Author Response

Dear Reviewer

Manuscript ID: sustainability-2257229

The paper titled " Does an environmental management system affect green innovation: The role of green financing in China’s tourism sector in a circular economy" has been reviewed as per the comments of the referee(s). These changes are incorporated and highlighted in the text; the details of the changes are as follows:

Reviewer 2 Comments

Reviewer 2 Changes

The English remains poor.

Demographic variables cannot take part of correlation analysis. Please remove them and redo the analysis

As per suggestion in the first round of revision proof reading of the manuscript is conducted from the experts of MDPI.

These changes are incorporated as per the suggestions of the reviewer by removing demographic variable from the analysis of correlation and make changes in the table and result section of the manuscript. These changes are highlighted in the manuscript.

Round 3

Reviewer 3 Report

Ok